# Intelligent Dynamic Spectrum Resource Management Based on Sensing Data in Space-Time and Frequency Domain

**DOI:** 10.3390/s21165261

**Published:** 2021-08-04

**Authors:** Deok-Won Yun, Won-Cheol Lee

**Affiliations:** 1Department of Electronic Engineering, Soongsil University, Seoul 06978, Korea; dhtor@naver.com; 2School of Electronic Engineering, Soongsil University, Seoul 06978, Korea

**Keywords:** spectrum resource management, cognitive radio, learning engine, reasoning engine, optimization engine

## Abstract

Edge computing offers a promising paradigm for implementing the industrial Internet of things (IIoT) by offloading intensive computing tasks from resource constrained machine type devices to powerful edge servers. However, efficient spectrum resource management is required to meet the quality of service requirements of various applications, taking into account the limited spectrum resources, batteries, and the characteristics of available spectrum fluctuations. Therefore, this study proposes intelligent dynamic spectrum resource management consisting of learning engines that select optimal backup channels based on history data, reasoning engines that infer idle channels based on backup channel lists, and transmission parameter optimization engines based genetic algorithm using interference analysis in time, space and frequency domains. The performance of the proposed intelligent dynamic spectrum resource management was evaluated in terms of the spectrum efficiency, number of spectrum handoff, latency, energy consumption, and link maintenance probability according to the backup channel selection technique and the number of IoT devices and the use of transmission parameters optimized for each traffic environment. The results demonstrate that the proposed method is superior to existing spectrum resource management functions.

## 1. Introduction

Based on the recent development of the Internet of things (IoT), a paradigm shift in computing technology for effectively processing the huge amounts of data generated by various IoT devices is rapidly occurring [1]. According to the International Data Corporation, it is predicted that by 2025, there will be 44.1 billion Internet connected IoT devices generating close to 79.4 zettabytes of data [2], representing a significant change in major end-end terminals from smart devices to the machines, sensors, and cameras that constitute the IoT. Gartner predicted that approximately 10% of enterprise generated data are already being processed outside of centralized data centers or the cloud, and that this number will reach approximately 75% by 2025 [3]. This suggests that cloud computing with a centralized computing structure has limitations in terms of accommodating various IoT services, such as smart factories, smart farms, and autonomous vehicles, that require real time processing. Therefore, because of the risk of overloading cloud servers based on the exponentially increasing data volume and network traffic, the transition of computing from existing remote clouds to network edges within the radio access network is expected to become a hot topic as a new paradigm [4,5,6]. Edge computing is a distributed computing paradigm that brings computation and data storage closer to the sources of data. This means that when computing services are processed in a location close to the terminal device used by the user, the user can receive faster and more stable services, and can benefit from flexible hybrid cloud computing. In particular, this method is effective for overcoming the problem of long latency, which is necessary for real time data processing and decision making in smart factories using the IoT. It also supports real-time decision making through sensing data collection and analysis.

However, the limited spectrum resources of edge computing cause frequent spectrum handoff in dynamic spectrum environments that require machine type device (MTD) operation in real time, and serious performance degradation may occur due to interference with other homogeneous networks. Therefore, it should be dynamically optimized according to time varying parameters such as channel state information. To this end, in previous studies [7,8,9], the usage patterns of incumbent users were learned using the occupancy probability and state transition probability based on history data representing spectrum statistics to infer idle channels in a dynamic spectrum environment. In addition, in [10,11,12,13], in order to enable continuous communication without interference to incumbent users, a reactive handoff method based on spectrum sensing and a proactive handoff method using a backup channel list are defined. However, when incumbent users exist in adjacent channel and location, the performance of incumbent users may be degraded due to the influence of interference, which causes a decrease in the spectrum efficiency and an increase in the number of spectrum handoff due to the restriction on the use of idle channels.

In this regard, recent studies [14] adopted rate splitting multiple access (RSMA) to support the massive access of IoT devices and achieved desirable performance in terms of interference suppression, spectrum efficiency and hardware complexity. In addition, interference management techniques such as beam forming [15] and resource allocation [16] methods were applied to solve interference problems between sub-networks, and furthermore, through the deep reinforcement lerning (DRL) learning framework, minimized interference between heterogeneous 5G cells [17]. However, Since these tasks rely on the general assumption that there is a central node with perfect knowledge of all parameters [18,19], the actual implementation may be constrained when considering the huge cost of security issues and overhead. Therefore, a distributed optimization approach that individually optimizes spectrum resources based on sensing information for each network is required [20,21,22].

In this paper, we propose an intelligent dynamic spectrum resource management structure based on sensing data that can coexist with other cognitive radio (CR) networks in adjacent channel and location while efficiently using limited spectrum resources. As shown in Figure 1, the proposed sensing data based intelligent dynamic spectrum resource management structure constructs an infrastructure based CR network in which a base station using the TV white space(TVWS) band controls IoT devices in a suburban area, where broadband access support is difficult, according to the IEEE 802.22 wireless regional area network (WRAN) standard [23,24]. At this time, Each base station constituting a single wireless network independently manages spectrum resources in the CR network through database access for self coexistence [25] between cells of the same type existing in adjacent channel and location. The spectrum resource management function consists of learning, reasoning and optimization engines. First, the learning engine generates a list of backup channels through characterization, spectrum determination, and classification based on history data representing the statistical information of the existing local network to facilitate timely switching to idle channels. A reasoning engine using the backup channel list performs a spectrum handoff operation to another idle channel to enable continuous communication without interfering with incumbent users. At this time, efficient spectrum resource management that enables self coexistence between adjacent networks is achieved through an optimization engine when incumbent users existing in adjacent channel and location are affected by interference.

This paper introduces the technology applied for the learning, reasoning, and optimization techniques that constitute the sensing-data-based intelligent dynamic spectrum resource management. In addition, through simulation, the intelligent dynamic spectrum resource management performance is analyzed according to the backup channel selection techniques and the number of IoT devices and the use of transmission parameters optimized for each traffic environment in time, space, and frequency domains.

## 2. Learning Engine

### 2.1. Incumbent User Activity Modeling Based on an On/Off Model

In this study, the channel status was periodically checked through spectrum sensing in the time and frequency domains, and the results were stored in a database. We also assumed that there was a CR master (CM) that can deliver channel information available to CR users (CUs) based on the collected channel information.

The data representing the statistical characteristics of incumbent users in the time and frequency domains use an On/Off model, which is represented by the incumbent user activity model in the CR network [26]. The On state indicates that a CU cannot use the channel because an incumbent user is currently using the channel; the Off state indicates that the incumbent user has no activity, so the CU can use the channel opportunistically.

The occupancy state probability for indicating the On and Off states for each channel is defined by the following equation, where $\lambda_{on}^{j}$ and $\lambda_{off}^{j}$ denote the average values of the On and Off state holding times of the *j*-th channel, respectively and each channel has a different exponential distribution characteristic:(1)$$P_{on}^{j}=\frac{\lambda_{on}^{j}}{\lambda_{on}^{j}+\lambda_{off}^{j}}.$$
$P_{on}^{j}$, which is calculated based on the average values of $\lambda_{on}^{j}$ and $\lambda_{onff}^{j}$ for the On and Off state holding times, indicates the probability that an incumbent user is in the On or Off state at every time slot in the *j*-th channel. To represent the On and Off states based on this probability value, the conditional expression below was adopted in this study.

The *k* value of this conditional expression corresponds to a random number between zero and one that is randomly generated for each channel during a given time $T_{i}$. At this time, if the random number *k* in the *j*-th channel at time $T_{i}$ is greater than the occupied state probability, then it is judged that the incumbent user is highly likely to be Off and the function returns a zero state. Conversely, if the occupied state probability is high, then the incumbent user in the *j*-th channel is likely to be On and the function returns a one state.
(2)$$C_{j}\left(T_{i}\right)=\left\{\begin{array}{cc} 0, & P_{on}^{j}\leq k\\ 1, & P_{on}^{j}>k \end{array}\right.$$

By using such a conditional expression, as shown in Figure 2, it is possible to define the activity patterns of incumbent users using zeroes and ones, which represent the state information of the incumbent user from the past to the present. These history data support a backup channel selection mechanism that can be characterized by occupancy and state transition probabilities.

### 2.2. Backup Channel Selection Technique Based on History Data

Identifying an appropriate channel that can continue to transmit data is the most urgent problem in spectrum mobility research [27] and frequent spectrum handoff caused by incorrect channel selection may degrade overall performance. The most common approach to solving this problem is to use a list of backup channels defined in the IEEE 802.22 WRAN standard [23]. A backup channel list is a list of channels classified according to their quality. This list is defined in advance so that users can switch to an idle channel to avoid having an interference effect on incumbent users. As for the channel selection method for generating such a backup channel list, occupancy probability and state transition probability based channel selection methods have been studied based on history data representing spectrum statistics [7,8,9].

The occupancy probability based backup channel selection technique using history data calculates the occupancy probability $P_{on}^{j}$ by counting the time slots in which the occupancy status $C_{j}\left(T_{i}\right)$ of the incumbent users is one among the total number of time slots for each channel.
(3)$$P_{on}^{j}=\frac{1}{M}\sum_{i=1}^{m}C_{j}\left(T_{i}\right)$$

The occupancy probabilities calculated for each channel are arranged in descending order to generate the backup channel list. The highest-priority channel is the channel with the lowest frequency of occurrence of incumbent users among all time slots.

The state transition probability based backup channel selection technique is defined with a state value of one when two consecutive time slots in the *j*-th channel are idle as indicated in the following conditional expression for minimizing spectrum handoff operations caused by incumbent users when reasoning an idle channel. Otherwise, the state value is set to zero. This enables the selection of a channel that can be used continuously without handoff operations, even if the channel has the same occupancy probability as another channel.
(4)$$F_{j}\left(T_{i},T_{i+1}\right)=\left\{\begin{array}{cc} 1, & \mathrm{if}\;T_{i}\;\mathrm{and}\;T_{i+1}\mathrm{are}\;{}^{\prime }\;\mathrm{off}\;{\mathrm{state}}^{\prime }\\ 0, & \mathrm{otherwise}, \end{array}\right.$$
where i∈M, M=1,2,3,...,m.
(5)$$P_{j}\left(T_{i},T_{i+1}\right)=\frac{\sum_{i=1}^{m-1}F_{j}\left(T_{i},T_{i+1}\right)}{m-1}$$

The state transition probability is calculated by counting the time slots with $F_{j}\left(T_{i},T_{i+1}\right)$ equal to one during the total time slot section characterized by the conditions defined above. Thereafter, The state transition probabilities calculated for each channel are sorted in descending order and the lowest state transition probability value is assigned the highest priority. at this time, The channel corresponding to the highest priority is the channel with the fewest handoff operations.

However, due to the imperfect pulse shaping of the interfering source transmitters in adjacent channel and location, as well as unwanted emission components generated by the individual elements constituting transmitters, the prior method may have an interference effect on incumbent users. Therefore, in this paper, we propose a channel selection method based on the two state transition probability (TSTP) that can minimize handoff and promote self coexistence with incumbent users in both the time and frequency domains.

TSTP based channel selection uses the following conditional expression for each time and frequency domain for the goals of self coexistence with incumbent users and minimizing the number of handoff: First, if both time slots are in an idle state (zero) or when they are converted from one to zero, then the probability of switching to an idle state is defined as one. Otherwise, it is defined as zero. The history data characterized in time units by the above conditional expression are defined by values of one when both adjacent channels in the frequency domain indicate an idle state. Otherwise, they are defined by values of zero.
(6)$$F_{j}\left(T_{i},T_{i+1}\right)=\left\{\begin{array}{cc} 1, & \mathrm{if}\;T_{i}\;\mathrm{and}\;T_{i+1}\mathrm{are}\;{}^{\prime }\;\mathrm{off}\;{\mathrm{state}}^{\prime }\\ 1, & \mathrm{if}\;T_{i}\mathrm{is}\;{}^{\prime }\;\mathrm{on}\;{\mathrm{state}}^{\prime }\mathrm{and}\;T_{i+1}\mathrm{is}\;{}^{\prime }\;\mathrm{off}\;{\mathrm{state}}^{\prime }\\ 0, & \mathrm{otherwise}, \end{array}\right.$$
where i∈M, M=1,2,3,...,m.
(7)$$F_{j}\left(C_{j-1},C_{j+1}\right)=\left\{\begin{array}{cc} 1, & \mathrm{if}\;C_{j-1}\mathrm{and}\;C_{j+1}\mathrm{are}\;1\\ 0, & \mathrm{otherwise}, \end{array}\right.$$
where j∈N, N=1,2,3,...,n.

The TSTP is calculated by counting the time slots in which $F_{j}$ is one in Equation (Equation 5) during the period of m−1 time slots characterized under the conditions defined above. The list of backup channels sorted in descending order according to the calculated TSTP values facilitates the preferential selection of channels that do not have incumbent users in adjacent channel in traffic scenarios with a large number of idle channels. This not only minimizes handoff and promotes self-coexistence with incumbent users, but also minimizes the transmission parameter optimization process and enables rapid optimal radio resource allocation.

## 3. Reasoning Engine

### Spectrum Handoff Techniques

When a spectrum is shared opportunistically by identifying an idle channel defined as an unused spectrum hole or white space at a specific time and location, it must not interfere with incumbent users. Therefore, if an incumbent user is detected during the idle channel reasoning process, then the operation should be paused and switched to another idle channel to maintain communication. This switching method is defined as a spectrum handoff, and the spectrum handoff strategy is typically divided into reactive and proactive handoff [12,13].

The reactive handoff enables accurate handoff operations on other idle channels by spectrum sensing when an incumbent user is detected, as shown in Figure 3. However, high handoff latency occur because the method only determines whether incumbent users are present by randomly selecting all channels without considering history data representing the usage patterns of past incumbent users.

The proactive handoff method predicts the arrival of an incumbent user based on a list of backup channels using history data and performs a handoff operation to another idle channel in advance, as shown in Figure 4 Accordingly, the handoff latency is shorter than that of the reactive handoff method. However, in a dynamic spectrum environment, there is a risk of wasting scarce frequency resources based on erroneous reasoning regarding the arrival of incumbent users.

Therefore, in this paper, we propose a reactive handoff process based on a list of backup channels using history data in advance. As shown in Figure 5, the reactive handoff process based on a backup channel list first detects an idle channel via spectrum sensing and then determines whether data are transmitted. At this time, if there is an incumbent user, a handoff to the channel corresponding to the next priority in the backup channel list is performed to determine whether data are transmitted through idle channel detection in the same manner as described above. Conversely, in the case of an idle channel, the channel is continuously used until an incumbent user is detected by monitoring every predetermined time on the selected channel. This minimizes the energy consumption caused by unnecessary spectrum handoff operations and helps solve the frequency resource problem by selecting correct idle channels using spectrum sensing.

## 4. Optimization Engine

As mentioned previously, when an incumbent user appears, the CU detects an idle channel and performs a handoff to another idle channel to avoid interference with the incumbent user. However, if incumbent users exist in adjacent channel and location, the performance of incumbent users may be degraded due to the effects of radio wave interference, which causes frequent spectrum handoff as a result of restrictions on the use of idle channels. Therefore, in this paper, we propose a transmission parameter optimization technique that can guarantee resource saving and service quality using interference analysis and genetic algorithm techniques based on limited radio resources to improve the spectrum efficiency of idle channels.

### 4.1. Monte Carlo Algorithm Based on Interference Analysis

In this paper, we propose a Monte carlo algorithm based interference analysis for transmission parameter optimization. Interference analysis based on the Monte carlo algorithm is largely divided into user interfaces, event generation engines, and interference calculation engines [28].

First, the user interfaces defines the parameters (antenna height, transmission power, center frequency, etc.) for the victim in the interference environment scenario, as well as a propagation loss model according to specific distance, location, and topography features. The interference environment is divided into victim and interfering links, as shown in Figure 6. The victim links refer to incumbent users in use at a particular location and time in the licensing band. An interfering link refers to any communication that temporarily uses an idle channel that is not used by the incumbent user at a specific time and region through the opportunistic frequency use method based on CR technology in the licensed band.

The event engine calculates the desired received signal strength (dRSS)and the interfered received signal strength (iRSS) through repeated simulations of *N* events according to the frequency separation and space distance. First, dRSS defines the transmit power $\left(P_{w}\right)$ and antenna gain ($g_{w\to v}{,}_{v\to w}$) for the wanted transmitter and victim receiver constituting the victim link, and uses a path loss model that considers the distance between the wanted transmitter and the victim receiver in the city center, as well as the propagation environment, which is defined by a number of unspecified obstacles.
(8)$$dRSS=P_{w}+g_{w\to v}-PL_{w\to v}+g_{v\to w}$$

$iRSS_{unwanted,i}$ is calculated based on the imperfect pulse shaping of the interfering source transmitter and propagation environment defined by the unwanted emission component $\left(emission(f_{It},f_{Vr})\right)$ of the interfering source, as shown in Figure 7. This component appears in individual elements constituting the transmitter and a number of unspecified obstacles between the interfering transmitter and interfering receiver. Additionally, to obtain the sum of received signal powers from *n* interfering transmitters operating in different channel and random locations, $iRSS_{unwanted,i}$ in dBm is used as an exponent and then converted into a logarithm to be returned, as defined in Equation (Equation 10).
(9)$$iRSS_{unwanted,i}=emission\left(f_{It},f_{Vr}\right)+g_{i\to v}-PL_{i\to v}+g_{v\to i}$$
where i∈N, N=1,2,3,...,n.
(10)$$iRSS_{unwanted,i}=10\log_{10}\left(\sum_{i=1}^{n}10^{\frac{iRSS_{unwanted,i}}{10}}\right)$$

Finally, according to the process of Figure 8, the interference engine calculates $C/I_{trial}$ using the iRSS value generated from the event with a dRSS value greater than the sensitivity and determines “Good” or “Interfered” through comparisons to the interference protection ratio of the victim receiver. This process is repeated until the end of events, and after the event are terminated, the interference probability of $N_{interfered}$ for which interference occurred among the total number of events is finally derived.

### 4.2. Elite Strategy Based on Genetic Algorithm

The radio resources obtained through Monte carlo algorithm based interference analysis are used as inputs for an optimal radio resource reconstruction method based on a genetic algorithm and The process of deriving the optimal transmission parameters according to the service goals of the incumbent user is performed [29].

The genetic algorithm used for optimal radio resource reconstruction is an optimization algorithm that mimics the natural evolutionary process through biological modeling. According to Darwin’s theory of evolution, weak species that cannot adapt to the environment are eliminated and strong species are characterized by traits with a high probability of being passed on to the next generation. The genetic algorithm process consists of initial population generation, fitness evaluation, and basic genetic operators (reproduction, crossbreeding, and mutation). The definition and operations of each process are discussed below [30].

The initial population representing the group in the first generation is defined as a set of *N* chromosome individuals with a binary string value.
(11)$$P\left(k\right)=s_{1}\left(K\right),s_{2}\left(k\right),s_{3}\left(k\right),...,s_{n}\left(k\right)$$
where $s_{i}\left(k\right)$, which represents the *i*th chromosome, is a point in the search space and the initial population $\left(P(k=0)\right)$ was generated using a random initialization method that initializes chromosomes with NL (i.e., group size chromosome length) binary integers.

In the fitness evaluation, a decoding process is first performed on each chromosome in the population and By substituting an input variable *x* converted into a phenotype into the objective function F(x), the fitness of the input variable for the solution is calculated by the fitness function. At this time, The fitness function must always be described in the form of a maximization problem and must not have a negative value. Accordingly, in this study, the following fitness function was defined to add an appropriate constant γ to the objective function F(x) prior to inversion:(12)$$f\left(x\right)=\frac{1}{F(x)+\gamma }.$$
where γ is a constant that satisfies the relationship F(x)+γ>0 for all input variables *x*. This constant ensures that f(x) does not have an excessively large value at the minimum value of F(x). However, because the selection pressure drops when γ is fixed, regardless of the generation progress, an optimal input variable was adaptively applied as generations progressed through the fitness scaling window. As discussed in the fitness evaluation process, the scaling window uses a sufficiently small value based on experience and experimental results because it is difficult to determine γ in advance in a real environment. However, if γ is fixed, regardless of the generation progress, the problem occurs where the selection pressure decreases in later generations.

Therefore, with scaling window, it is possible to maintain a consistent selection pressure by continuously changing γ to the minimum value of the objective function in the past multiple generation population. When the number of populations used is the scaling window $\left(W_{s}\right)$, there are three scaling methods that can be used depending on the size of the scaling window as shown in Table 1.

Reproduction, crossover, and mutation operations are performed sequentially following the fitness calculation. The first reproduction is performed by selecting entities in the population P(k) based on the fitness values to form a mating pool P(k+1). This selection allows the genes in a population to spread widely through the population of subsequent generations by eliminating weak chromosomes in the population and selecting strong chromosomes. Methods for implementing the reproduction algorithm include roulette wheel selection, tournament selection, and rank based selection. In this study, the roulette wheel selection algorithm, which is widely used in genetic algorithms, was used. The roulette wheel selection algorithm selects and duplicates chromosomes from the previous population according to the magnitudes of their selection probabilities. However, because genes cannot be altered, it does not affect the overall genotype of the population. Additionally, based on the probabilistic properties of this algorithm, the disadvantage is that it may not be possible to select the optimal chromosome during the selection process. Therefore, the elite strategy was also adopted to supplement roulette wheel selection. This strategy ensures that the strongest chromosome in the group is guaranteed to be duplicated to the next generation without extinction.

Crossover randomly selects a parent chromosome pair from the mating pool and generates offspring by exchanging and combining bits after a crossover point. This operation is repeated until the size of the new generation is equal to the size of the parent population. Methods for implementing crossover include one point crossover, multi-point crossover, cycle crossover, and uniform crossover. In this study, offspring were generated utilizing the one point crossover method. The one point crossover method is often called standard crossbreeding and is a basic operator in genetic algorithms. The operations during one cycle can be divided into three stages as shown in Table 2.

Through reproduction and crossover, the population gradually evolves into chromosomes with similar shapes at the end of each generation. However, at the beginning of the next generation, the population can fall into a semi optimal solution or dead corner based on a lack of gene diversity. Mutation is used as a method to overcome this problem.

Mutation plays the role of preventing certain bits in all chromosomes from becoming fixed in early generations by changing the bits in chromosomes based on a mutation probability. This process is divided into the three stages as shown in Table 3.

## 5. Simulations

### 5.1. Scenario

In this study, for reasoning idle channels based on a list of backup channels and optimizing transmission parameters to facilitate coexistence with incumbent users in adjacent channel and location, as shown in Figure 9, a simulation environment that is dynamically variable in the time, space, and frequency domains was designed in three steps.

As a first step, using the On/Off model, history data that is dynamically variable by time and frequency domains is based on the superframe structure defined in IEEE 802.22 Mac [31]. The history data representing the superframe consists of 16 time slots with a length of 10 ms for each globally synchronized time slot, and 55 individual orthogonal subchannels according to the domestic DTV broadcasting standard. At this time, each has a bandwidth of 100 kHz within the bandwidth occupied by 5.5 MHz except for the 250 kHz guard bands on both sides of the 6 MHz channel bandwidth defined in ATSC broadcasting [32]. And, by using the history data configured in units of superframes as described above, the number and distribution of incumbent users in the space domain are determined according to the channel usage status in units of time slots in [33,34].

The second step of forming *n* single wireless networks in the space domain is deploying wireless networks that each opportunistically service IoT devices in random locations within a 5 × 5 km square area [35] using channels representing the incumbent user occupancy statuses of the 55 individual orthogonal sub channels in the *m*-th time slot. At this time, in order to derive the optimal transmission parameter of the interfering transmitter that satisfies the standard of −6dB for the I/N protection ratio of the victim receiver at the minimum receiving power position through interference analysis [36], the IoT assumes the worst case situation located from the service boundary (600 m) from a base station [37]. Table 4 shows the victim link parameters for the simulation.

As a final step, Figure 10 depicts a situation where 255 IoTs [31] perform uplink operation at any location within the service area (600 m) of a base station located in the center of a 5 × 5 km square area. This indicates that unwanted emission of massive IoT devices performing uplink operation with a maximum transmission power of 12.6 dBm per 100 kHz [32] at a random location can have a serious radio wave interference effect on incumbent users located in adjacent channel and cell boundaries [38,39]. In addition, in this study, the Extended Hata propagation model [28] and the unwanted emission mask of the wireless microphone service [40] were applied to reflect the propagation loss due to an unspecified obstacle between the interfering source and the receiving unit. Table 5 and Table 6 shows the interfering link parameters and unwanted emission mask for the simulation.

### 5.2. Objective Function

In this study, the interferer transmit power consists of chromosomes with a string length of 11 bits and precision within one decimal place (*d*: 1) within the search interval from 0 to 12.6 dBm.
(13)$$l\geq \log_{2}\left[\left(x_{U}-x_{L}\right)10^{d}+1\right]$$
Subsequently, chromosomes with 11 bit string lengths were converted into interfering transmitter power values through a decoding process, and we analyzed the interference effect on *n* victim receivers distributed at random locations within a 5 × 5 km square area during 100 events per input parameters. At this time, the objective function used in this study is defined below for deriving the parameter that minimizes the sum of the squared errors between the probabilities of interference generated during 100 events according to the I/N interference protection ratio standard for each chromosome and the 5% interference probability reference values applied for field testing [41,42].
(14)$$F\left(x\right)=\sum_{i=1}^{n}\sqrt{{\left(P_{required}-P_{trial}^{i}\right)}^{2}}$$

Then, by selecting strong chromosomes through fitness evaluation and elite strategy based reproduction, crossover, and mutation operations, the genes in the initial population spread widely through the populations in subsequent generations. Table 7 lists the main parameter setting values required for the calculation process for *N* generations.

### 5.3. Performance Evaluation Method

In this study, to analyze spectrum resource management performance according to the use of history data and the proposed backup channel selection technique, the number of spectrum handoff, handoff latency, link maintenance probability, energy consumption, and spectrum efficiency were verified [13].

First, in the idle channel reasoning process [12,43], the number of spectrum handoff is defined as the number of times that data transmission is stopped and switched to another idle channel according to incumbent user detection during one session of CR data transmission. However, frequent spectrum handoff operations can degrade the communication performance based on latency and increased energy consumption. Therefore, this simulation aimed to minimize unnecessary spectrum handoff to realize an efficient CR network. The number of spectrum handoff operations according to incumbent user detection was counted for each time slot, as defined in the following equation:(15)$$F\left(T_{i}\right)=\left\{\begin{array}{cc} 1, & \mathrm{if}\;T_{i}\mathrm{is}\;{}^{\prime }\;\mathrm{on}\;{\mathrm{state}}^{\prime }\\ 0, & \mathrm{otherwise}, \end{array}\right.$$
where i∈M, M=1,2,3,...,m.
(16)$$Count\left(T_{i}\right)=\sum_{i=1}^{m}F\left(T_{i}\right)$$

The spectrum handoff latency [12,44] represents the number of channel searches that occur during the handoff process to an idle channel according to the detection of an incumbent user. To verify a channel selection technique that can quickly handoff to an idle channel with a small number of searches during the reasoning period, as shown in Equations (Equation 17) and (Equation 18), when an incumbent user is detected in each time slot, the number of channel searches occurring during the handoff process is counted according to the backup channel order.
(17)$$F\left(C_{ij}\right)=\left\{\begin{array}{cc} 1, & \mathrm{if}\;C_{ij}\mathrm{is}\;{}^{\prime }\;\mathrm{on}\;{\mathrm{state}}^{\prime }\\ 0, & \mathrm{otherwise}, \end{array}\right.$$
where i∈N, N=1,2,3,...,n.
(18)$$Count\left(C_{ij}\right)=\sum_{i=1}^{m}\sum_{j=1}^{n}F\left(C_{ij}\right)$$

In a temporally and spatially dynamic CR network, frequent spectrum handoff by incumbent users can degrade IoT communication performance as a result of excessive sensing energy consumption [45]. Therefore, in this study, as shown in Equations (Equation 19) and (Equation 20), the reduction in sensing energy consumption was confirmed by counting the number of times that the same idle channel was maintained in a continuous time interval without a spectrum handoff operation over a total of *m* time slots.
(19)$$F\left(T_{i},T_{i+1}\right)=\left\{\begin{array}{cc} 1, & \mathrm{if}\;T_{i}\mathrm{and}\;T_{i+1}\mathrm{is}\;{}^{\prime }\;\mathrm{off}\;{\mathrm{state}}^{\prime }\\ 0, & \mathrm{otherwise}\; \end{array}\right.$$
(20)$$Count\left(T_{i},T_{i+1}\right)=\sum_{i=1}^{m-1}F\left(T_{i},T_{i+1}\right)$$

A pause in communication most likely occurs when an idle channel does not exist at a specific time in a traffic environment with a large number of incumbent users. Additionally, radio wave interference caused by the unwanted emission of an interference source limits the use of available idle channels [16,43]. Therefore, in this study, to evaluate spectrum management performance for continuous communications in such a scenario, the number of time slots in which a communication link was successfully maintained up to *T* was counted.
(21)$$F\left(T_{i}\right)=\left\{\begin{array}{cc} 1, & \mathrm{if}\;T_{i}\mathrm{is}\;{}^{\prime }\;\mathrm{CU}\;\mathrm{on}\;{\mathrm{state}}^{\prime }\\ 0, & \mathrm{otherwise}\; \end{array}\right.$$
(22)$$Count\left(T_{i}\right)=\sum_{i=1}^{m}F\left(T_{i}\right)$$

Radio wave interference caused by the unwanted emission of an interfering transmitter may limit the use of available idle channels, resulting in a scenario in which transmission is temporarily suspended during the spectrum handoff process, which significantly affects the number and handoff latency and increases energy consumption. Therefore, in this study, spectrum efficiency [46] changes were analyzed based on the number of available idle channels satisfying the threshold of 5% or less interference probability according to whether the proposed transmission parameter optimization was applied.
(23)$$P_{\mathrm{spectrum}\;\mathrm{efficiency}\;}=\frac{1}{M}\sum_{i=1}^{m}\left(\frac{\mathrm{Number}\;\mathrm{of}\;\mathrm{realistic}\;\mathrm{channels}\;\mathrm{in}\;\mathrm{the}\;i-\mathrm{th}\;\mathrm{time}\;\mathrm{slot}}{\mathrm{Number}\;\mathrm{of}\;\mathrm{idle}\;\mathrm{channels}\;\mathrm{at}\;\mathrm{the}\;i-\mathrm{th}\;\mathrm{time}\;\mathrm{slot}}\right).$$

### 5.4. Performance Evaluation Results

#### 5.4.1. Backup Channel List Performance Comparative Analysis for Channel Selection

We identified the most efficient channel selection method through a comparative analysis of spectrum resource management performance by using a list of backup channels for each channel selection method prior to applying optimization via Figure 11.

If a backup channel list sorted in random channel order is used without considering the past activity patterns of incumbent users, then the probability of selecting a suboptimal channel is very high. This eventually causes a handoff operation in every time slot, resulting in an average of 6.5 or more handoff in a 0.2 traffic environment with multiple idle channels and one can seen that channels are used continuously with an average number of 8.5 time slots. Additionally, In addition, it is difficult to maintain a backup channel list as the number of channel searches that increases by approximately 100 or more during handoff to idle channels whenever traffic increases. This causes a pause at specific time slots, and eventually, the handoff increase gradually decreases from 0.3 traffic, and at the same time, the link maintenance performance is 2% lower than that of other channel selection techniques. In contrast, in the case of a channel selection method based on the occupancy probability and state transition probability using history data representing past incumbent user activity patterns, channels with low activity can be preferentially selected during the idle channel reasoning process. This makes the same channels continuously available with fewer than 70 lower channel searches and an average of nine time slots. Additionally, by maintaining the backup channel list up to 0.3 traffic, we confirmed continuous handoff operations and a link maintenance probability of at least 95%.

The TSTP proposed in this study considers the possibility that two consecutive time slots will be idle or occupied. Additionally, in a traffic scenario with a large number of idle channels, a channel without an incumbent user in an adjacent channel is selected preferentially. As a result, the average number of time slots that can be used continuously at 0.2 traffic was confirmed to be 9.8. By maintaining a list of backup channels with approximately 100 fewer channel searches than the random channel selection technique, we determined that it is possible to perform a quick handoff operation with a link maintenance probability of more than 95% up to 0.3 traffic.

#### 5.4.2. Spectrum Resource Management Performance Analysis Depending on Whether or Not Optimization Is Applied

We compared and analyzed the performance of the spectrum resource management technique depending on whether optimization was applied during the handoff operation based on the list of backup channels using TSTP.

In Figure 12, the green graph shows that when there are 255 IoT devices that perform uplinks with maximum transmission power (12.6 dBm) on a total of 46 idle channels, radio interference restricts the use of 18 idle channel channels adjacent to incumbent users. on the other hand, When optimizing the transmit power of the interferer to satisfy the criterion of less than 5% interference probability in the 18 idle channels whose use is restricted by interference, the red bar graph indicates that the transmission power that can coexist with incumbent users is approximately 5 dBm in the nearest channel and 10 dBm in the second adjacent channel according to the frequency separation caused by the unwanted emission mask.

Figure 13 presents the average spectrum efficiency depending on whether optimization is applied or not each time slot, as defined in Figure 12 using 500 history data per traffic environment. Overall, the graph exponentially decreases as the traffic gradually increases. At this time, if optimization is applied, the spectrum efficiency is the most improved from 0.3 traffic to approximately 5.7% by the use of idle channels whose use was restricted by interference. Even above 0.3 traffic, the spectrum efficiency for idle channels is significantly improved by optimization. However, as traffic increases, the number of available idle channels gradually decreases, so the increasing trend clearly slows.

In Figure 14, the spectrum efficiency improvement according to the optimization application shows the largest increase/decrease in handoff and energy consumption performance at 0.3 traffic, and has a link maintenance performance close to 100%. In addition, the number of channel searches lowers from 0.7 traffic to about 70 or less and the continuous handoff operation confirms that the link maintenance performance improved by about 11% or more even when incumbent users were mixed. After 0.8 traffic, the number of available idle channels decreases and the increase trend decreases sharply.

#### 5.4.3. Spectrum Resource Management Performance Analysis According to the Number of IoT Devices

We analyzed the number of IoT devices and spectrum resource management performance with a link maintenance performance of 95% or more for each traffic based on a list of backup channels using TSTP and spectrum resource management with optimization applied.

Figure 15 presents the average spectrum efficiency according to the number of IoT devices for at least 10, 50, 100, and 255 devices when optimization is applied in every time slot, as defined in Figure 12 using 500 history data for each traffic environment. In general, the graph exhibits an exponential decrease as traffic gradually increases. however, as the number of devices decreases, the use of idle channels with limited use caused by interference increases, thereby significantly improving the spectrum efficiency.

In Figure 16, the case of spectrum resource management with 10 IoT devices, continuous handoff operations are possible to 0.8 traffic the average number of channels searched is less than 500, and the number of continuously available time slots increases by seven or more. As a result, we confirmed that a link maintenance performance of more than 95% is achievable at 0.8 traffic. For spectrum resource management using 50 IoT devices, continuous handoff operations were possible with fewer than 250 channel searches and an average of five consecutive time slots in 0.7 traffic, with a link maintenance performance of more than 95%. Finally, in the case of 100 IoT devices, continuous handoff operation was possible with less than 50 channel searches in 0.5 traffic and 2 consecutive time slots on average, and it was confirmed that the link maintenance performance was greater than 95%.

## 6. Conclusions

In this paper, we proposed an intelligent dynamic spectrum resource management structure based on sensing data for the efficient use of frequency self coexistence between single wireless networks opportunistically providing IoT services in the TVWS band. The proposed sensing data based intelligent dynamic spectrum resource management structure is divided into learning, reasoning, and optimization engine. Learning engine uses a history based TSTP channel selection method based on the On/Off model, and reasoning engine uses a reactive spectrum handoff technique based on a backup channel list. Transmission parameters are optimized by a genetic algorithm using Monte carlo algorithm based interference analysis.

Spectrum efficiency improvement through the use of the intelligent dynamic spectrum resource management technique based on sensing data facilitates continuous handoff operations by maintaining a backup channel list and facilitates continuous use of the same channels by guaranteeing a low number of channel searches when searching for idle channels. This significantly reduces the energy consumption of rapidly switching to idle channels and performing spectrum sensing. Additionally, reliable communication can be maintained in a traffic scenario mixed with incumbent users with massive IoT devices with a link maintenance performance of at least 95%.

In the future, we will apply optimal transmission parameters stored in a database to case by case classification and matching using machine learning and case based reasoning, which should reduce the computational load of the intelligent dynamic spectrum resource management technique and enable rapid solution delivery.

## Figures and Tables

**Figure 1 sensors-21-05261-f001:**
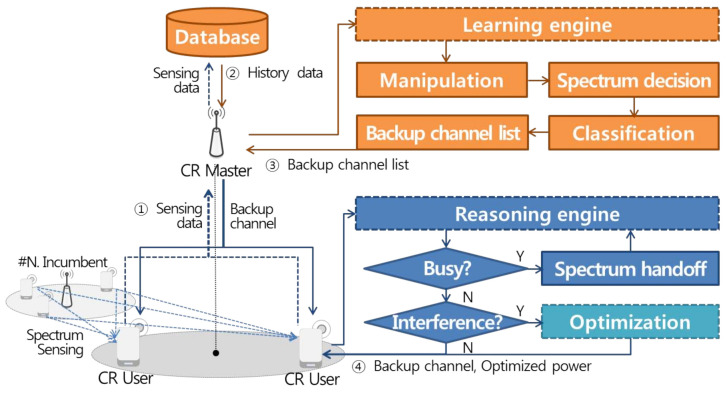
Sensing data based intelligent dynamic spectrum resource management structure.

**Figure 2 sensors-21-05261-f002:**
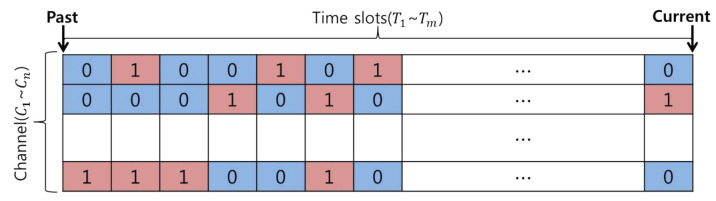
Incumbent user activity modeling based on an On/Off model.

**Figure 3 sensors-21-05261-f003:**
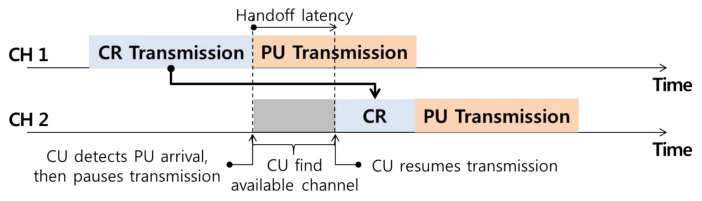
Reactive spectrum handoff technique.

**Figure 4 sensors-21-05261-f004:**
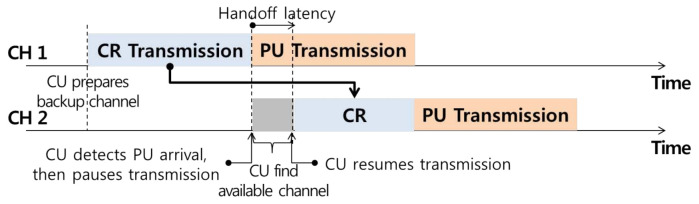
Proactive spectrum handoff technique.

**Figure 5 sensors-21-05261-f005:**
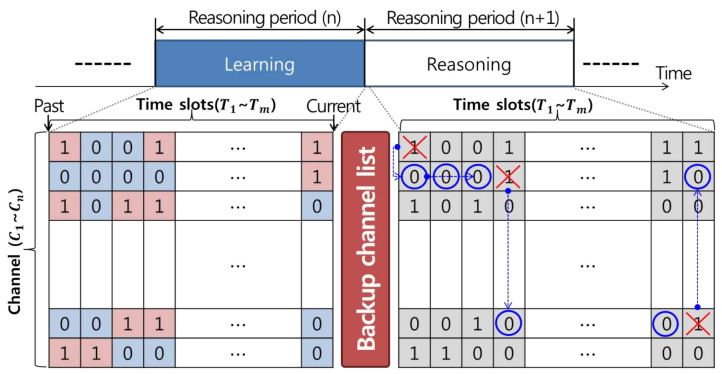
Reactive spectrum handoff technique using a backup channel list.

**Figure 6 sensors-21-05261-f006:**
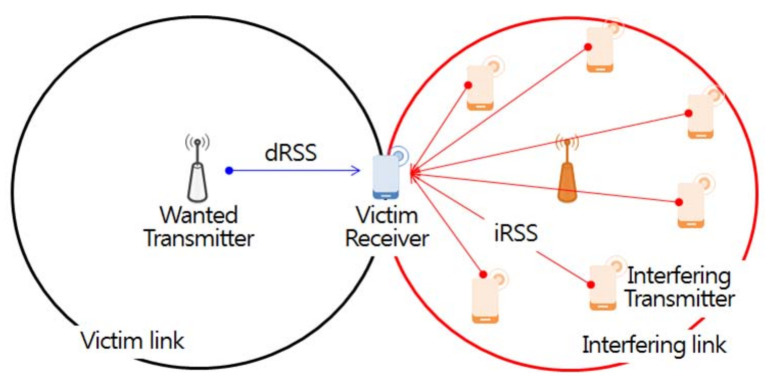
Victim link and interfering link.

**Figure 7 sensors-21-05261-f007:**
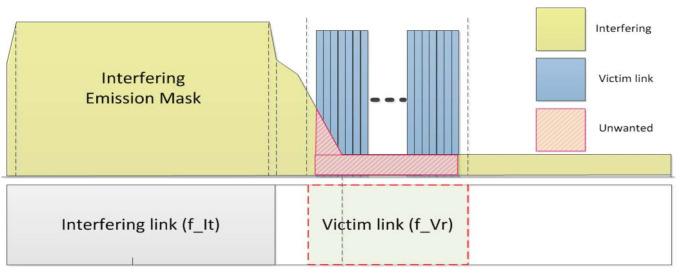
Interference caused by unwanted emission masks.

**Figure 8 sensors-21-05261-f008:**
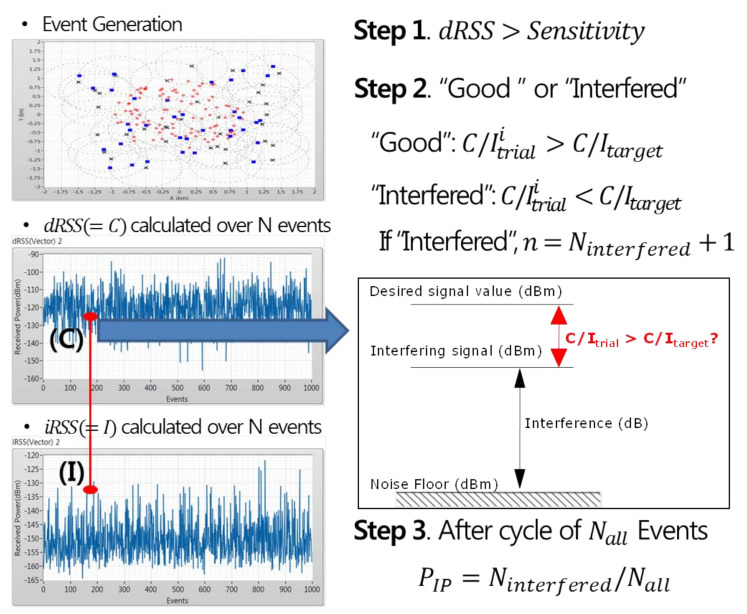
Interference probability calculation.

**Figure 9 sensors-21-05261-f009:**
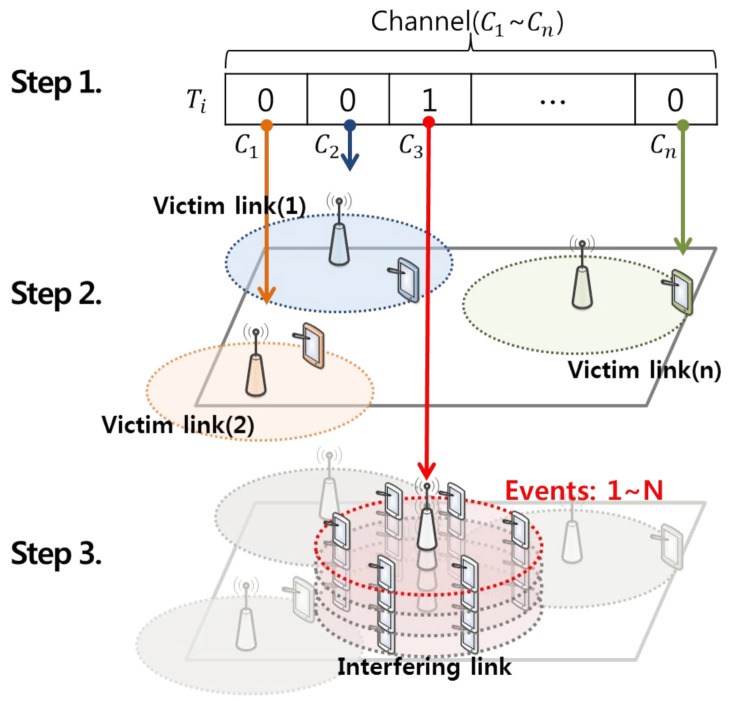
Setup of the simulation environment.

**Figure 10 sensors-21-05261-f010:**
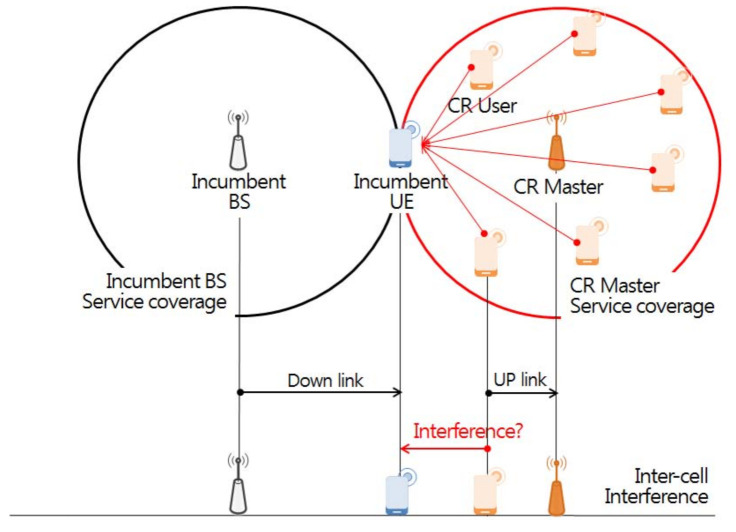
Inter cell interference scenario.

**Figure 11 sensors-21-05261-f011:**
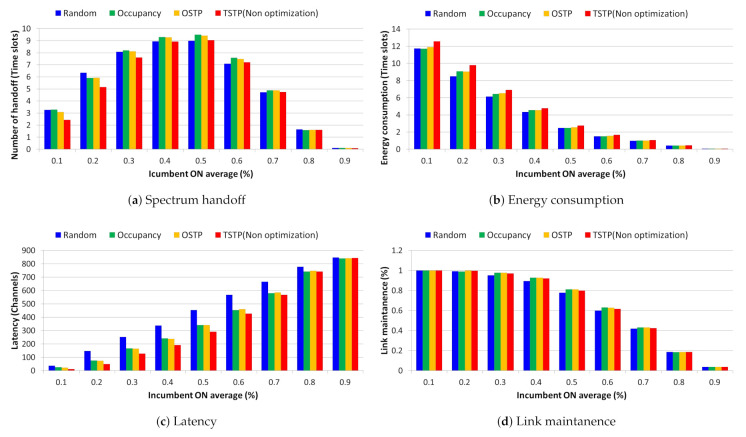
Backup channel list performance comparative analysis ((**a**) Spectrum handoff, (**b**) Energy consumption, (**c**) Latency, (**d**) Link maintenance) for each channel selection technique.

**Figure 12 sensors-21-05261-f012:**
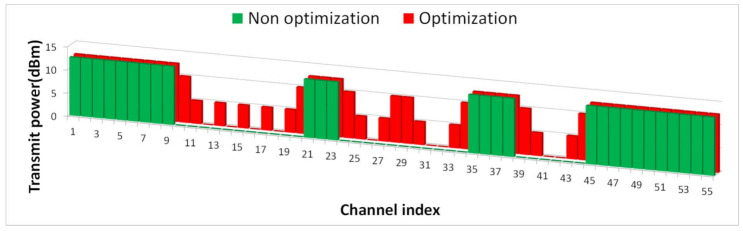
Numbers of idle channels depending on whether or not optimization is applied.

**Figure 13 sensors-21-05261-f013:**
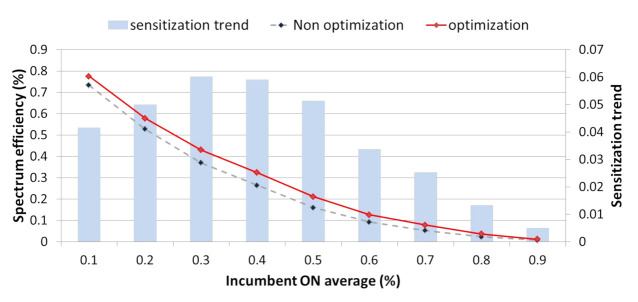
Comparative analysis of spectrum efficiency depending on whether or not optimization is applied.

**Figure 14 sensors-21-05261-f014:**
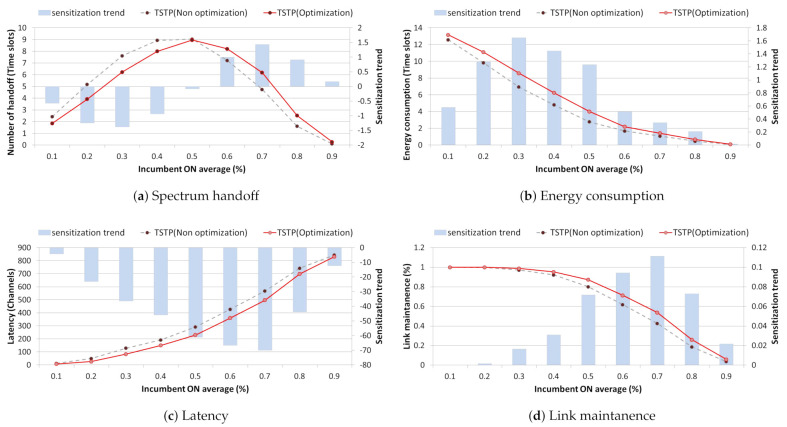
Comparative analysis of spectrum resource management performance ((**a**) Spectrum handoff, (**b**) Energy consumption, (**c**) Latency, (**d**) Link maintenance) depending on whether or not optimization is applied.

**Figure 15 sensors-21-05261-f015:**
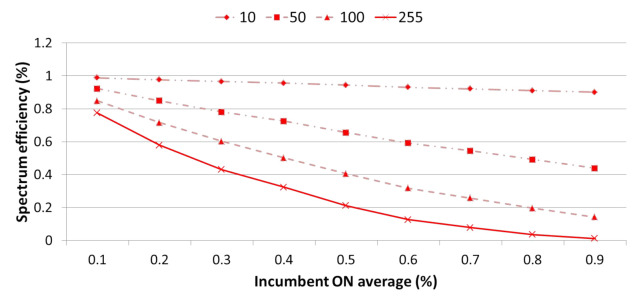
Comparative analysis of spectrum efficiency according to the number of IoT devices.

**Figure 16 sensors-21-05261-f016:**
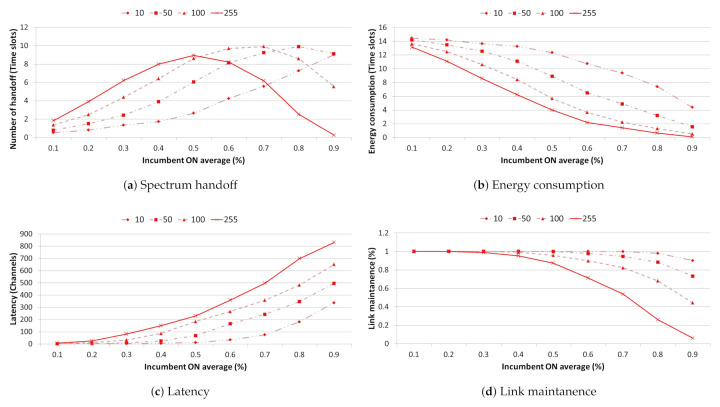
Comparative analysis of spectrum resource management performance ((**a**) Spectrum handoff, (**b**) Energy consumption, (**c**) Latency, (**d**) Link maintenance) according to the number of IoT devices.

**Table 1 sensors-21-05261-t001:** Three scaling methods depending on the size of the scaling window.

$W_{s}$ = 0	In the first generation, γ is initialized to the minimum value of the objective function. In subsequent generations, it changes only when the objective function values of all objects are greater than γ.
0 < $W_{s}$ < $\gamma_{min}$	We change γ to the minimum value of the objective function found in the past generation $W_{s}$.
$W_{s}$ = $\gamma_{min}$	This means that the window size is infinite and no scaling is performed. Therefore, the initial γ value does not change and is used continuously.

**Table 2 sensors-21-05261-t002:** One-point crossover method.

Step 1	Random selection of parental chromosome pairs in a mating pool.
Step 2	Determine whether to perform crossover based on the mating probability $P_{c}$:
	- If a randomly generated random number γ∈[0,1] is $\gamma \leq P_{c}$, then two offspring are generated by selecting a random crossing point c∈[1,l−1] and a parent chromosome pair, and exchanging genes after the crossing point;
	- Conversely, in the case of $\gamma >P_{c}$, the selected parent chromosome pair is stored intact as offspring;
	- *l* denotes the chromosome length.
Step 3	The operations above are repeated until the generated offspring are filled with *N* chromosomes in the temporary population.

**Table 3 sensors-21-05261-t003:** Mutation method.

Step 1	Sequentially select one bit from the chromosomes in the population P(k+1).
Step 2	Determine whether or not the selected bit should be mutated based on the mutation probability $P_{m}$:
	- Generate a random number γ∈[0,1]. In the case of $\gamma \leq P_{m}$, if the randomly selected bit is one, then it is inverted to zero. If it is zero, then it is inverted to one;
	- Conversely, for $\gamma >P_{c}$, bit inversion does not occur.
Step 3	Selected bits are duplicated within the population P(k+1).

**Table 4 sensors-21-05261-t004:** Victim link parameters.

Parameters	Value
Victim receiver center frequency ($f_{Vr}$)	600.2 MHz
Antenna height ($H_{w,v}$)	1.5 m
Antenna gain ($g_{w\to v}$)	6 dBi
Noise floor ($N_{f}$)	−167.83 dBm
Bandwidth (*B*)	100 kHz
Protection ratio ($I/N_{required}$)	−6 dB

**Table 5 sensors-21-05261-t005:** Interfering link parameters.

Parameters	Value
Interfering transmitter center frequency ($f_{It}$)	Among 55 sub-channels with a bandwidth of 100 kHz, the idle channel selected based on the backup channel is defined as the center frequency.
transmit power ($P_{I}t$)	- Optimal transmission power that satisfies the criteria for interference probability within 5%;
	- (Min) 1 dBm (Max) 12.6 dBm.
Antenna height ($H_{i,v}$)	1.5 m
Antenna gain ($g_{i\to v}$)	6 dBi
Path loss ($PL_{i\to v}$)	Extended Hata model

**Table 6 sensors-21-05261-t006:** Break points of the unwanted emission mask (B: Bandwidth).

Frequency Relative to the Center of the Channel	Relative Level (dBc)
−1 MHz	−90
− B	−80
−0.5 B	−60
−0.35 B	−20
−0.25 B	0
0.25 B	0
0.35 B	−20
0.5 B	−60
B	−80
1 MHz	−90

**Table 7 sensors-21-05261-t007:** Genetic algorithm parameters.

Parameters	Value
Chromosome length	11 bit
Population size	15
Number of generations	100
Crossover probability	0.8
Mutation probability	0.01

## Data Availability

Not applicable.

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
