# Peer review of "Intelligent Dynamic Spectrum Resource Management Based on Sensing Data in Space-Time and Frequency Domain"

_sensors, 2021, doi:10.3390/s21165261_

Round 1
Reviewer 1 Report
Please see the attached file "Comments to sensors-1310313" for the detailed comments.

Reviewer 2 Report
This paper presents an interesting spectrum resource management technique through sensing data in space, time and frequency domains. The work is generally well-written and could be suitable for publication in the Sensors journal provided that the following comments are implemented within the document:
- A clear explanation regarding the novelty of the paper with respect to previous published works related to spectrum resource management should be better indicated in the Introduction section. Some more references should be added in this sense.
- The improvement of resource saving and service quality with respect to the case when the performance of incumbent users is degraded due to radiowave interferences should be somehow quantified.
- The 5% threshold criteria for interference probability should be properly justified.
- Please change the labels which appear in Korean in Figure 8.
- The entire manuscript should be revised by a professional proofreading service.
Reviewer 3 Report
This paper proposes a learning technique to characterize channel usage patterns for optimal backup channel selection towards efficient spectrum resource management.
The paper is well-written and, although a well-known and extensively investigated topic is considered, it gives some interesting insights for the dynamic spectrum utilization. Therefore, I suggest only some minor revisions.
- The introduction is well-written, but is can be enhanced by including a dedicated paragraph for related/relevant work on Spectrum Sensing and ML-based approaches for resource optimization. This would be achieved by adding some relevant studies, such as:
[1] Haykin, S., Thomson, D. J., & Reed, J. H. (2009). Spectrum sensing for cognitive radio. Proceedings of the IEEE, 97(5), 849-877.
[2] Giannopoulos, A., Spantideas, S., Tsinos, C., & Trakadas, P. (2021, June). Power Control in 5G Heterogeneous Cells Considering User Demands Using Deep Reinforcement Learning. In IFIP International Conference on Artificial Intelligence Applications and Innovations (pp. 95-105). Springer, Cham.
[3] Zhao, Q., & Sadler, B. M. (2007). A survey of dynamic spectrum access. IEEE signal processing magazine, 24(3), 79-89.
- Some clear statements are required to define the paper’s scope and contributions.
- Fig. 8 needs substantial revision due to bad quality and illustration.
- Figs 13 and 15 are extremely stretched.
- Reconsider carefully the sectioning and the titles of each section.
Round 2
Reviewer 1 Report
The authors have well addressed all my concerns in the last round of review.
It can be published in its current version after revising the following minor errors:
1) "accessin" of Ref.[14] should be revised as "access in";
2) "beamformingfor" of Ref.[15] should be revised as "beamforming for".